# Seasonal Monitoring Method for TN and TP Based on Airborne Hyperspectral Remote Sensing Images

Lei Dong [1,2], Cailan Gong [1,*], Xinhui Wang [3], Yang Wang [1], Daogang He [1,2], Yong Hu [1], Lan Li [1] and Zhe Yang [1,2]

1   Key Laboratory of Infrared System Detection and Imaging Technologies, Shanghai Institute of Technical Physics, Chinese Academy of Sciences, Shanghai 200083, China; fcdl@mail.ustc.edu.cn (L.D.)
2   University of Chinese Academy of Sciences, Beijing 100049, China
3   Key Laboratory of Spatial-Temporal Big Data Analysis and Application of Natural Resources in Megacities, Shanghai Municipal Institute of Surveying and Mapping, Shanghai 200063, China
*   Correspondence: gcl@mail.sitp.ac.cn

**Abstract:** Airborne sensing images harness the combined advantages of hyperspectral and high spatial resolution, offering precise monitoring methods for local-scale water quality parameters in small water bodies. This study employs airborne hyperspectral remote sensing image data to explore remote sensing estimation methods for total nitrogen (TN) and total phosphorus (TP) concentrations in Lake Dianshan, Yuandang, as well as its main inflow and outflow rivers. Our findings reveal the following: (1) Spectral bands between 700 and 750 nm show the highest correlation with TN and TP concentrations during the summer and autumn seasons. Spectral reflectance bands exhibit greater sensitivity to TN and TP concentrations compared to the winter and spring seasons. (2) Seasonal models developed using the Catboost method demonstrate significantly higher accuracy than other machine learning (ML) models. On the test set, the root mean square errors (RMSEs) are 0.6 mg/L for TN and 0.05 mg/L for TP concentrations, with average absolute percentage errors (MAPEs) of 23.77% and 25.14%, respectively. (3) Spatial distribution maps of the retrieved TN and TP concentrations indicate their dependence on exogenous inputs and close association with algal blooms. Higher TN and TP concentrations are observed near the inlet (Jishui Port), with reductions near the outlet (Lanlu Port), particularly for the TP concentration. Areas with intense algal blooms near shorelines generally exhibit higher TN and TP concentrations. This study offers valuable insights for processing small water bodies using airborne hyperspectral remote sensing images and provides reliable remote sensing techniques for lake water quality monitoring and management.

**Keywords:** airborne sensing; integrated learning; segmented modelling; TN; TP

## 1. Introduction

Nitrogen and phosphorus play vital roles as nutrients in aquatic ecosystems. However, when present in excessive amounts, they can cause eutrophication, leading to a cascade of environmental problems including algal blooms, fish kills, and habitat degradation for aquatic organisms [1]. In particular, small water bodies are more likely to experience eutrophication problems due to their relatively small volume and susceptibility to surrounding environmental influences [2–4]. That is why it is important to regularly monitor and control the total nitrogen (TN) and total phosphorus (TP) levels in water bodies to prevent eutrophication and maintain the ecological balance [5–9].

The traditional manual monitoring methods for total nitrogen (TN) and total phosphorus (TP) are time-consuming, labour-intensive, and provide only spot information [10–13]. The application of remote sensing has proven to be an effective approach for the monitoring of water quality parameters such as total nitrogen (TN) and total phosphorus (TP). Currently, the remote sensing data employed for water quality monitoring encompass satellite remote sensing, unmanned aerial vehicle (UAV) remote sensing, and large-scale manned aircraft remote sensing imagery. However, satellite remote sensing is subject to

certain limitations in inland water monitoring, including low spatial resolution, lengthy revisit periods, extensive spectral bands, atmospheric influences, and a low signal-to-noise ratio [2,6,10]. Unmanned aerial vehicle (UAV) remote sensing offers high spatial resolution and is advantageous for monitoring small watersheds, such as rivers. Nevertheless, it becomes impractical for larger areas due to factors such as endurance and the weather [8]. In comparison to satellite and UAV imagery, manned-aircraft-derived airborne remote sensing imagery is distinguished by a number of advantages. Manned aircraft imagery offers a higher spatial and spectral resolution, which enables the capture of finer details [14]. In comparison to unmanned aerial vehicles (UAVs), manned aircraft are capable of carrying larger payloads and flying for longer periods, which enables them to cover a wider area [15]. Consequently, the utilisation of the airborne remote sensing imagery obtained from manned aircraft is becoming increasingly prevalent in the study of inland small-to-medium-sized lakes and reservoirs [16].

TN and TP are non-optically active water quality parameters with weak spectral characteristics, which makes it challenging to accurately determine their concentrations using traditional linear or polynomial fitting methods [6,17,18]. The advent of machine learning (ML) methods has led to their emergence as a valuable tool for the monitoring of non-optically active water quality parameters, such as total nitrogen (TN) and total phosphorus (TP) [5,6,9,19–22]. At present, the most commonly employed machine learning algorithms for the retrieval of water quality parameters include regularised linear regression (LRR), random forest regression (RFR), kernel ridge regression (KRR), Gaussian process regression (GPR), and support vector machine regression (SVR) [6]. Previous studies have demonstrated that XGBoost and CatBoost are effective tools for water quality monitoring [23]. The objective of this study is to further explore the performance of ensemble learning algorithms in monitoring TN and TP in order to improve the accuracy and reliability of TN and TP monitoring. Previous studies have typically fitted inversion models for water quality parameters to data from all dates together [24–26]. However, this approach frequently fails to consider the subtle variations in water quality parameters across different seasons. Indeed, numerous studies have identified significant seasonal variations in water quality parameter concentrations [11]. Conversely, optically active water quality parameters such as the chlorophyll-a (chla) concentration exhibit greater seasonal fluctuations, as evidenced by various studies [24,27]. Consequently, some studies have employed data from the same season to establish inversion models, achieving more accurate results [27,28]. Nevertheless, previous research has also demonstrated that TN and TP undergo certain alterations in accordance with the seasonal changes [23]. In light of this observation, we decided to establish seasonal inversion models for TN and TP with the objective of improving our understanding of and ability to predict their seasonal trends.

In summary, the primary objectives of this study are as follows: (1) the establishment of seasonal retrieval models for TN and TP; (2) the identification of the most suitable method for fitting small sample sizes among the current mainstream ensemble learning algorithms; and (3) the inversion of TN and TP concentrations of lakes and rivers in the study area using airborne hyperspectral remote sensing imagery, with the aim of leveraging the advantages of airborne remote sensing imagery in inland small water body monitoring. The study demonstrated the feasibility of using airborne remote sensing images with high spatial resolution for water quality monitoring in small inland water bodies. The proposed seasonal monitoring model for TN and TP based on the CatBoost method was successfully applied in the study area, providing a valuable reference for inland water body management.

## 2. Data and Methods

### 2.1. Study Area

The study area is situated at the border between the Qingpu District of Shanghai Municipality and Kunshan City of Jiangsu Province, China. It encompasses Dianshan Lake, Yuandang Lake, and the principal inflow and outflow rivers of Dianshan Lake. Dianshan Lake is a significant body of water, acting as the receiving end of water for the Taihu Lake–Wujiang region

and as the source of the Huangpu River. In contrast, Yuandang Lake serves as a quasi-source water conservation area for the upper reaches of the Huangpu River, with multiple functions including regulating runoff, water supply, irrigation, navigation, and tourism.

### 2.2. Research Data

#### 2.2.1. Concentration Data

Continuous on-site measurements were conducted during the spring, summer, autumn, and winter seasons from 2018 to 2023 in multiple rivers and lakes in Shanghai Municipality, resulting in a total of 195 sets of valid data. The locations of the sampling points in Shanghai Municipality are shown in Figure 1b. Samples were collected from the water surface to a depth of 50 centimetres using a 2 L water sampler. To prevent sample deterioration, the collected water samples were placed in a thermos box maintained at a temperature of 0 °C to 6 °C, shielded from light, and promptly transported to the laboratory for freezing preservation. During the sampling process, care was taken to ensure that the sample bottles were filled to a sufficient level to minimise the interference of residual air on some analytical parameters [29]. The TN and TP contents were determined using a Hach DR3900 visible spectrophotometer in the laboratory. Previous studies have indicated that algal blooms frequently occur in Dianshan Lake during the summer and autumn seasons, with elevated concentrations of TN and TP observed during these two seasons. Therefore, the data were divided into two seasonal groups: the winter–spring season group (Dataset 1) and the summer–autumn season group (Dataset 2). The Table 1 presents the distribution of TN and TP concentrations in the two seasonal data groups, as well as the overall distribution of TN and TP concentrations in Dataset 0.

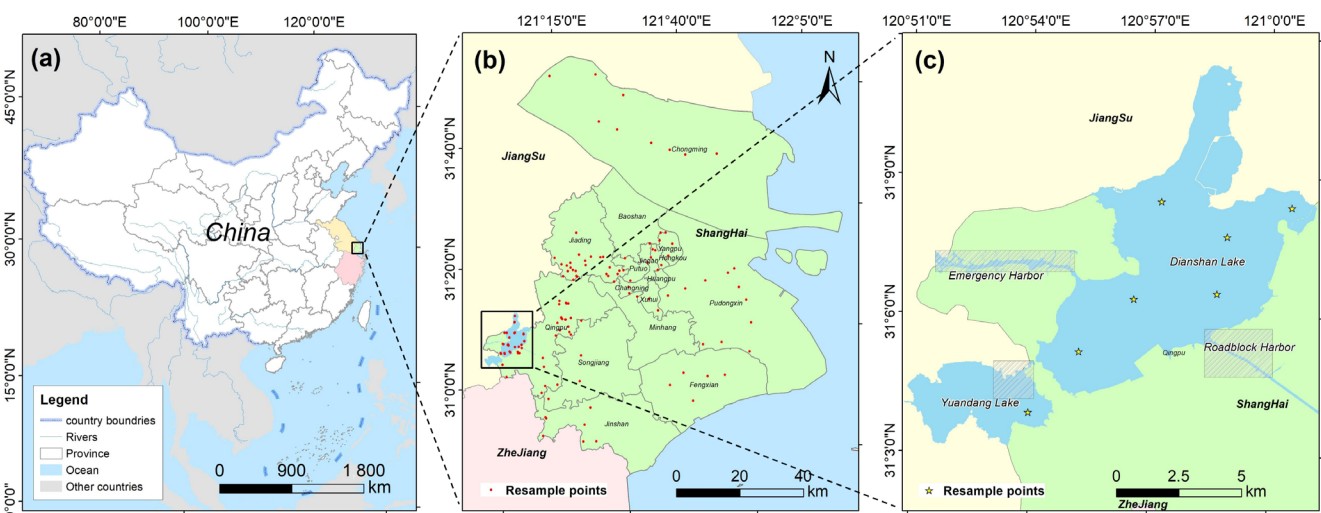

**Figure 1.** (**a**) Schematic map showing the location of the study area. (**b**) Schematic map of sampling points covering Shanghai Municipality from 2018 to 2023. (**c**) Schematic map of synchronised sampling points with UAV flight dates. See Appendix A, Figures A1 and A2, for enlargements of Figure 1b,c.

**Table 1.** Statistical description of measured water quality parameters.

| Dataset | Range (mg/L) | Mean ± Std (mg/L) | CV | N |
|---|---|---|---|---|
| Dataset 0 (TN) | 0.72–6.55 | 2.64 ± 1.13 | 0.43 | 195 |
| Dataset 1 (TN) | 0.72–5.66 | 2.51 ± 1.01 | 0.40 | 95 |
| Dataset 2 (TN) | 1.14–6.55 | 2.77 ± 1.23 | 0.45 | 100 |
| Dataset 0 (TP) | 0.041–0.664 | 0.163 ± 0.081 | 0.494 | 195 |
| Dataset 1 (TP) | 0.050–0.400 | 0.162 ± 0.066 | 0.409 | 95 |
| Dataset 2 (TP) | 0.041–0.664 | 0.164 ± 0.093 | 0.563 | 100 |

### 2.2.2. Reflectance Spectral Data

The collected water spectral data were synchronised using an ASD Fieldspec4 spectro-radiometer above the water surface, in accordance with the method proposed by Mobley et al. [30]. The reflectance spectra within the range of 400 nm to 907 nm were subsequently utilised for subsequent applications. These data include water-leaving radiance ($L_{sw}(\lambda)$), sky radiance ($L_{sky}(\lambda)$), and reference panel radiance ($L_p(\lambda)$). The water surface reflectance ($R_{rs}$) was calculated using the following formula.

$$R_{rs}(\lambda) = \frac{L_{sw}(\lambda) - \rho_{sky}(\lambda)L_{sky}(\lambda)}{\pi L_p(\lambda)/\rho_p(\lambda)},\qquad(1)$$

In the equation above, *r* represents the reflectance of sky-light at the water–air interface. Its value depends on the wind speed over the water surface. Since all experiments were conducted under clear and windless weather conditions, *r* is set to 0.028 in this study. $\rho_{sky}$ represents the sky window reflectance at the air–water interface. $\rho_p$ represents the radiance reflectance of the gray panel (with a reflectance of 30%). A total of five to ten measurements were conducted at each site, with the results subsequently averaged in order to enhance the reliability and representativeness of the findings.

The reflectance (Rrs) data, derived from field measurements, is presented in Figure 2. The reflectance data were divided into two groups according to the season of collection and the concentrations of TN and TP were compared between the two groups using bar graphs. The results demonstrate that the reflectance distribution range of Dataset 1 is significantly narrower, with an overall lower average reflectance compared to Dataset 2. It is noteworthy that the reflectance peaks and troughs of Dataset 1 are biased towards shorter wavelengths compared to Dataset 2, particularly in the red to near-infrared bands. The concentration bar graphs indicate that the TN concentration range in Dataset 1 is higher than Dataset 2, while the TP concentration range is lower than Dataset 2.

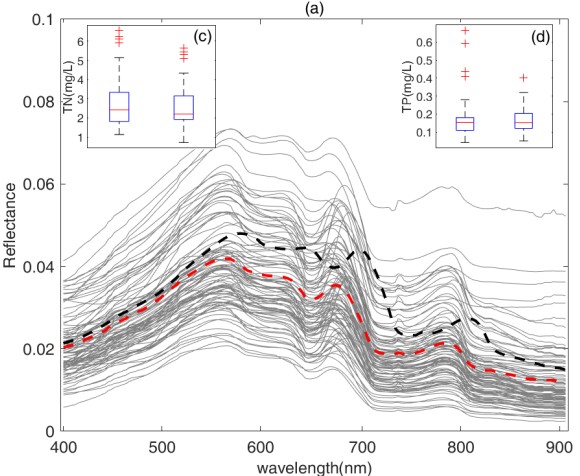 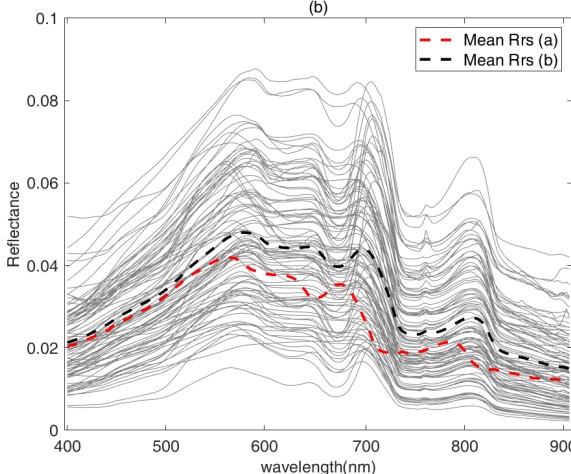

**Figure 2.** Field Measurements. (**a**) Reflectance spectral curves of Dataset 1 and (**b**) Dataset 2. The red dashed line represents the average Rrs of Dataset 1, while the black dashed line represents the average Rrs of Dataset 2. Bar graphs showing the concentrations of (**c**) TN and (**d**) TP. The first column represents Dataset 1, and the second column represents Dataset 2.

### 2.2.3. Airborne Hyperspectral Remote Sensing Data

From 15 to 17 June 2022, aerial photography of lakes and rivers in the study area was conducted using a manned aircraft equipped with an airborne multispectral and multimodal imaging spectrometer (AMMIS), resulting in the acquisition of high-resolution image data [31]. The performance parameters of AMMIS are presented in Table 2.

**Table 2.** Performance Parameters of AMMIS.

| Name | Indicator Parameters |
|---|---|
| Band | 0.4~0.95 μm |
| Number of Spectral Bands | $\geq$256 |
| Spectral Resolution (nm) | $\leq$5 |
| Spatial Resolution (m) | 0.75 m |

The raw data obtained from the aerial hyperspectral imaging spectrometer must be preprocessed in order to obtain the desired files. Firstly, due to the inability to directly apply the raw data obtained from AMMIS, which includes signal errors from the imaging system, the acquired metadata is uncompressed. This process involves subtracting dark current in order to obtain pixel brightness data along with flight auxiliary data such as GPS trajectory and POS posture. Subsequently, image processing is carried out in the following order: spectral calibration, radiometric calibration, geometric correction, and atmospheric correction. The study employs the gas emission spectral lamp method for spectral calibration. Visible and near-infrared spectral calibration utilises a mercury lamp to illuminate a standard white diffuse reflectance panel, capturing the image of the panel. The spectral offset is determined and corrected based on the position of the mercury lamp characteristic spectral lines in the pixels. As there exists a linear relationship between image DN values and radiance values, radiometric calibration of the image data is achieved by solving for the gain coefficients and offsets of various bands in the hyperspectral image. Following geometric correction and atmospheric correction, the normalised difference water index (NDWI) method will be employed to extract water areas for further analysis and application [32]. The calculation formula for *NDWI* is as follows:

$$NDWI = \frac{Rrs(\lambda_1) - Rrs(\lambda_2)}{Rrs(\lambda_1) + Rrs(\lambda_2)} \tag{2}$$

where $Rrs(\lambda_1)$ represents the green band and $Rrs(\lambda_2)$ represents the near-infrared band. In this study, bands 37 (555.63 nm) and 199 (864.27 nm) were selected. Pixels with an NDWI greater than 0 are determined as water bodies.

*2.3. Research Methodology*

2.3.1. Ensemble Learning Methodology

AdaBoost

AdaBoost is a classic ensemble learning method that constructs a robust learner by iteratively training a series of weak learners and adjusting the weight of each learner. In each iteration, AdaBoost focuses on the samples that were misclassified in the previous round, increasing their weight to pay more attention to difficult-to-classify samples. Through continuous iteration, AdaBoost can gradually improve the performance and generalisation ability of the model [33,34].

CatBoost

CatBoost is an ensemble learning method based on gradient boosting specifically designed to tackle tasks such as classification, regression, and ranking. Compared to traditional gradient boosting algorithms, CatBoost has unique advantages in handling categorical features and missing values. It can directly handle categorical features and missing values without the need for additional preprocessing steps, thereby simplifying the process of model construction. Furthermore, CatBoost utilises techniques such as the symmetric leaf node algorithm and adaptive learning rate, which enhance the performance and robustness of the model [24,35–38].

Random Forest

The random forest method is an ensemble learning approach based on decision trees. It enhances the performance of the model by constructing multiple decision trees and averaging or voting their prediction results. Random Forest is typically characterised by high robustness and interpretability, making it an appropriate choice for problems that require high interpretability and stability [22,23,38–40].

XGBoost

XGBoost is another gradient-boosting-based ensemble learning method that exhibits significant advantages in both performance and efficiency. XGBoost iteratively trains a series of decision trees by optimising the gradient of the loss function and combines them into a strong learner using boosting. XGBoost performs well in handling large-scale datasets and complex problems and typically provides higher prediction accuracy and better generalisation ability. [10,39,41–44].

### 2.3.2. Segmented Modelling

The approach taken involves dividing the data into two groups based on seasons: winter–spring and summer–autumn. For each dataset, suitable feature spectral bands or combinations are identified through the relationship between the measured spectra and TN and TP in order to establish retrieval models for TN and TP. The Python programming language is employed in this study for the grid search technique to determine the hyperparameters of the model. Each training process includes a five-fold cross-validation strategy to comprehensively evaluate the performance of the model. This segmented modelling approach differs from previous methods, yet it is believed to be more effective in capturing the characteristic changes of TN and TP in different seasons. During prediction, the data to be predicted is first classified into the corresponding seasonal group based on the date, then the corresponding inversion model is used for prediction. The objective of this method is to achieve more accurate predictions for the concentration changes of TN and TP during the frequent occurrence of algal blooms in summer and autumn. This will provide more precise data support for the management of Dianshan Lake water quality.

### 2.3.3. Accuracy Assessment

Dataset 1 and Dataset 2 are divided into training, validation, and testing sets in a ratio of 7:2:1. It is essential that the selected sites for the training, validation, and testing sets encompass the entire range of TN and TP concentrations. To ensure the applicability of the model, an in situ test set for spectral-to-image validation was created by including TN and TP concentration data measured on the day of hyperspectral satellite data collection. The evaluation criteria include the root mean square error (*RMSE*), mean absolute percentage error (*MAPE*), and bias. The formulas for each evaluation criterion are as follows:

$$RMSE = \sqrt{\frac{\sum_{i=1}^{N}(y_i - \hat{y}_i)^2}{N}} \tag{3}$$

$$MAPE = \frac{1}{N}\sum_{i=1}^{N}\left|\frac{y_i - \hat{y}_i}{y_i}\right| \times 100\% \tag{4}$$

$$Bias = \frac{1}{N}\sum_{i=1}^{N}|y_i - \hat{y}_i| \tag{5}$$

where $N$ represents the sample size, $y_i$ is the value of the i-th observed data point, and $\hat{y}_i$ is the value of the *i*-th predicted data point.

## 3. Results

### 3.1. ML Feature Selection

A Pearson-correlation-coefficient-based correlation analysis was conducted to investigate the correlation between water quality parameters and water surface reflectance. The Pearson correlation coefficient (r) ranges from −1 to +1, where values close to +1 indicate a strong positive correlation, values close to −1 indicate a strong negative correlation, and values close to 0 indicate little to no linear correlation. The correlation coefficients between Dataset 0, Dataset 1, or Dataset 2 and TN and TP were analysed, and it was found, as shown in Figure 3, that TN exhibited the highest correlation at the red edge band (700–750 nm). Furthermore, two significant correlation peaks were observed between TN and Dataset 2 in the green band (550 nm) and near-infrared band (beyond 800 nm). The peak in the green band was notably higher than that in the near-infrared band. Additionally, there were minor correlation peaks at the blue band (480 nm) and red band (655 nm). However, it is notable that TN exhibited a low correlation coefficient trough at 680 nm across all datasets. Similarly to TN, the correlation coefficient plot for TP in Figure 3 displays a comparable trend, with the exception that the correlation of TP at 700–850 nm was considerably higher than in the visible light bands.

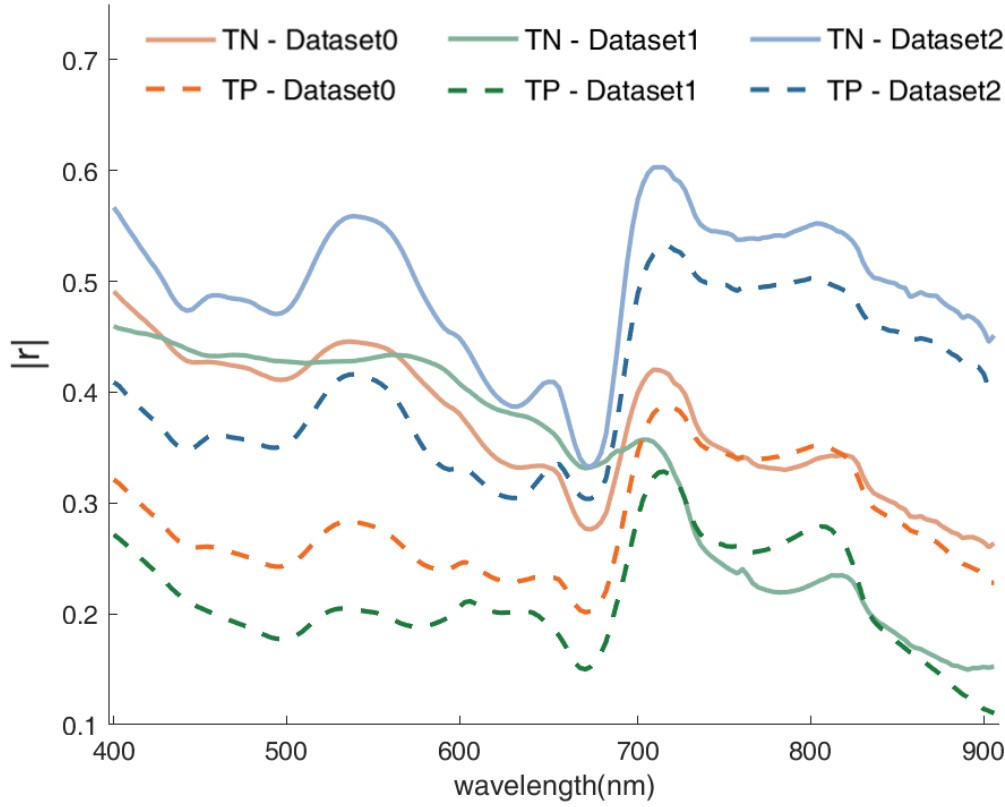

**Figure 3.** Relationship between the absolute values of Pearson correlation coefficients (r) of water surface reflectance and TN and TP.

The selection of features in machine learning is becoming increasingly important, as it helps to improve the model's ability to generalise and avoid overfitting by removing redundant information and selecting features that are highly correlated with the target variable but uncorrelated with each other. In this study, we have developed an innovative approach to fully utilise the rich information contained in the high-spectral-resolution data that we employed. Specifically, the spectral range of 400 to 900 nanometres was divided into six equally spaced intervals, each spanning 100 nanometres. The initial spectral bands within each interval were then identified. This novel method facilitates the more effective utilisation of information within high-spectral-resolution imagery, thereby enhancing the

richness and interpretability of the data. It also reduces computational costs, minimises redundancy, and improves the model's generalisation capabilities. It is anticipated that the adoption of this method will enhance the reliability and effectiveness of our research. Additionally, several common band combination methods were calculated: band differencing, band ratios, and band normalisation. These band combinations demonstrate robustness in water quality parameter retrieval and are therefore widely used by researchers. By exhaustively searching for all possible band combinations and employing a correlation-based feature selection method (CFS) [45], the four most effective band combination methods were selected as input features for the ML model. CFS is a feature selection technique based on correlation. Initially, it selects features with the highest correlation to the target variable (usually the prediction target) from all features as part of the initial subset. This ensures that the initial subset contains the most relevant features, providing a solid starting point for the model. Subsequently, the "advantage" of the feature subset is calculated by computing the interrelationships among features in the subset and the average correlation between these features and the target variable. Thereafter, CFS iteratively selects and adds features to the subset until the merit value can no longer be further improved. Table 3 below lists the selected features for the TN and TP retrieval models.

**Table 3.** Input features selected for the TN and TP retrieval models by the ML model.

| TN | | TP | |
|---|---|---|---|
| **Spring and Winter** | **Summer and Autumn** | **Spring and Winter** | **Summer and Autumn** |
| B(1) | B(1) | B(1) | B(1) |
| B(15) | B(21) | B(15) | B(21) |
| B(54) | B(47) | B(69) | B(46) |
| B(78) | B(84) | B(79) | B(86) |
| B(102) | B(103) | B(106) | B(107) |
| B(138) | B(135) | B(136) | B(135) |
| B(117)–B(118) | B(111)–B(121) | B(121)–B(145) | B(106) − B(14) |
| B(115)–B(118) | B(5)/B(6) | B(105)/B(59) | (B(85) − B(80))/(B(85) + B(80)) |
| B(118)/B(124) | B(111)–B(120) | B(122)–B(145) | B(105) − B(14) |
| B(114)–B(119) | B(84)/B(78) | B(105)/B(58) | B(85)/B(80) |

*3.2. Performance of Ensemble Learning Algorithms*

Four ML models were developed for TN and TP. During the training process for these models, they all utilised the same training dataset and determined hyperparameters through a grid search strategy. These hyperparameters were initially set based on the performance of the training set and then fine-tuned based on the evaluation metrics of the validation set. The accuracy results of the four models on the training set are shown in Table 4. In terms of both TN and TP, CatBoost demonstrated the most favourable results, followed by XGBoost, RF, and AdaBoost, which exhibited the least favourable results.

**Table 4.** The performance of the model on the training set.

| Parameters | Model | $R^2$ | RMSE | MAPE (%) | Bias | Slope |
|---|---|---|---|---|---|---|
| | AdaBoost | 0.42 | 0.94 | 29.49 | 0.69 | 1.08 |
| TN (mg/L) | Catboost | 0.89 | 0.52 | 14.97 | 0.36 | 1.03 |
| | RF | 0.53 | 0.86 | 28.52 | 0.66 | 1.08 |
| | XGBoost | 0.76 | 0.67 | 21.9 | 0.50 | 1.08 |
| | AdaBoost | 0.42 | 0.063 | 38.41 | 0.049 | 1.24 |
| TP (mg/L) | Catboost | 0.81 | 0.041 | 21.88 | 0.030 | 1.09 |
| | RF | 0.46 | 0.061 | 31.61 | 0.044 | 1.11 |
| | XGBoost | 0.57 | 0.056 | 26.62 | 0.038 | 1.13 |

The scatter plot (Figures 4 and 5) illustrates that TN is relatively evenly distributed within the range 1~6 mg/L. However, the results of the TN predictions demonstrate that all four methods tend to underestimate high concentrations. The predictions from the CatBoost method are more densely clustered around the 1:1 line, indicating higher accuracy and precision. The test set revealed that TP is distributed between 0 and 0.4 mg/L, with only one data point exceeding this range. All four methods underestimated this value, but the predictions from CatBoost exhibited the highest accuracy.

In order to evaluate the adaptability and accuracy of the models from the reflectance end to the image end, data from seven points in Dianshan Lake and Yuandang Lake were utilised that were measured synchronously on the same day using hyperspectral images as validation points. The sampling point locations are shown in Figure 1c. Four ML models were employed to calculate the concentrations of TN and TP. The following figure presents a comparison between the predicted values obtained from reflectance at the image end and the measured TN and TP concentrations.

Figure 6 illustrates that the CatBoost model produces predictions that are closely aligned with the measured values. This indicates that the Catboost model can be employed for the prediction of large-scale phenomena from satellite images.

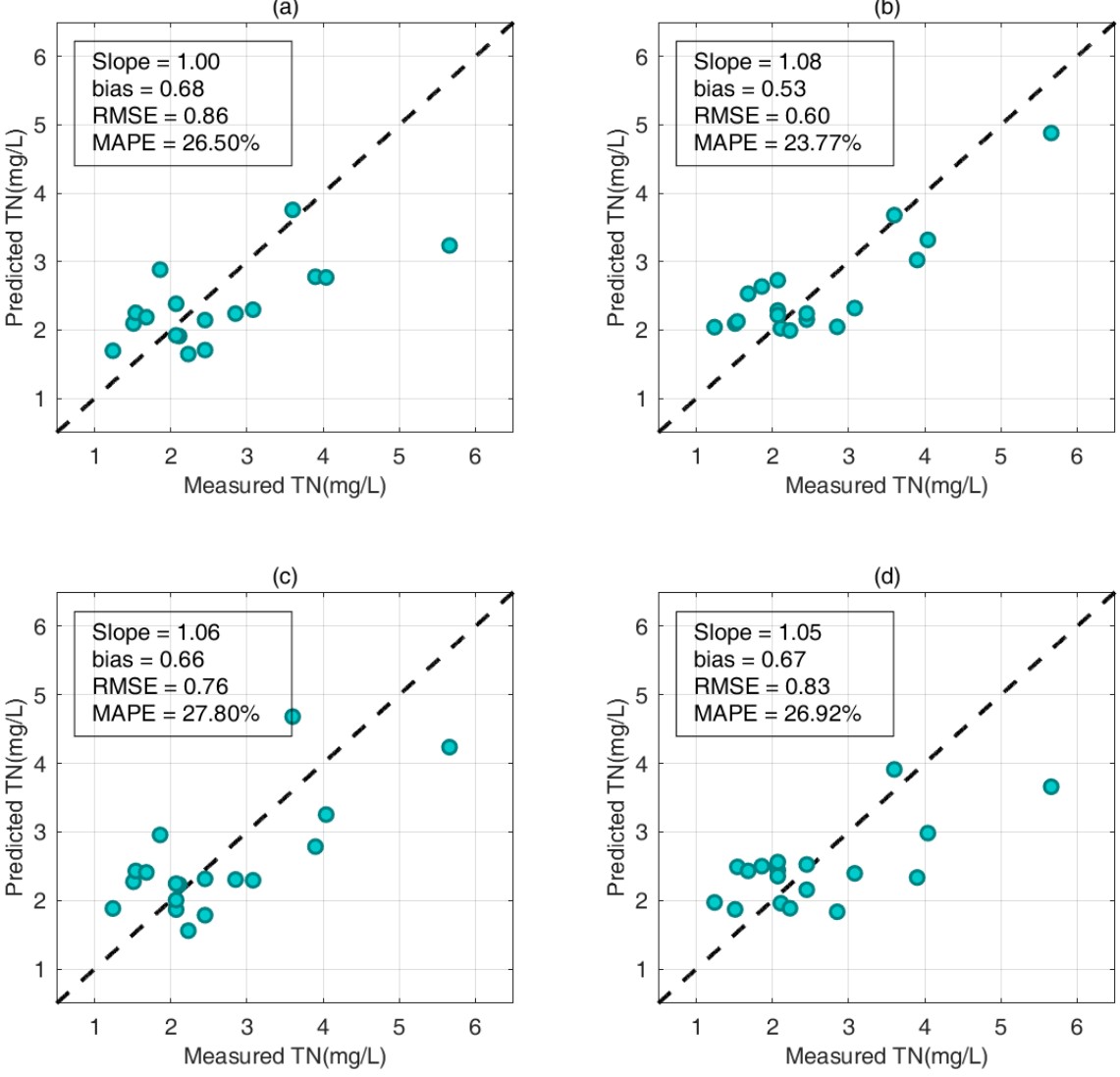

**Figure 4.** The scatter plot results of the predicted TN concentrations on the test set by four ensemble learning algorithms (AdaBoost (**a**), Catboost (**b**), RF (**c**), and XGBoost (**d**)), where the black dashed line represents the 1:1 line.

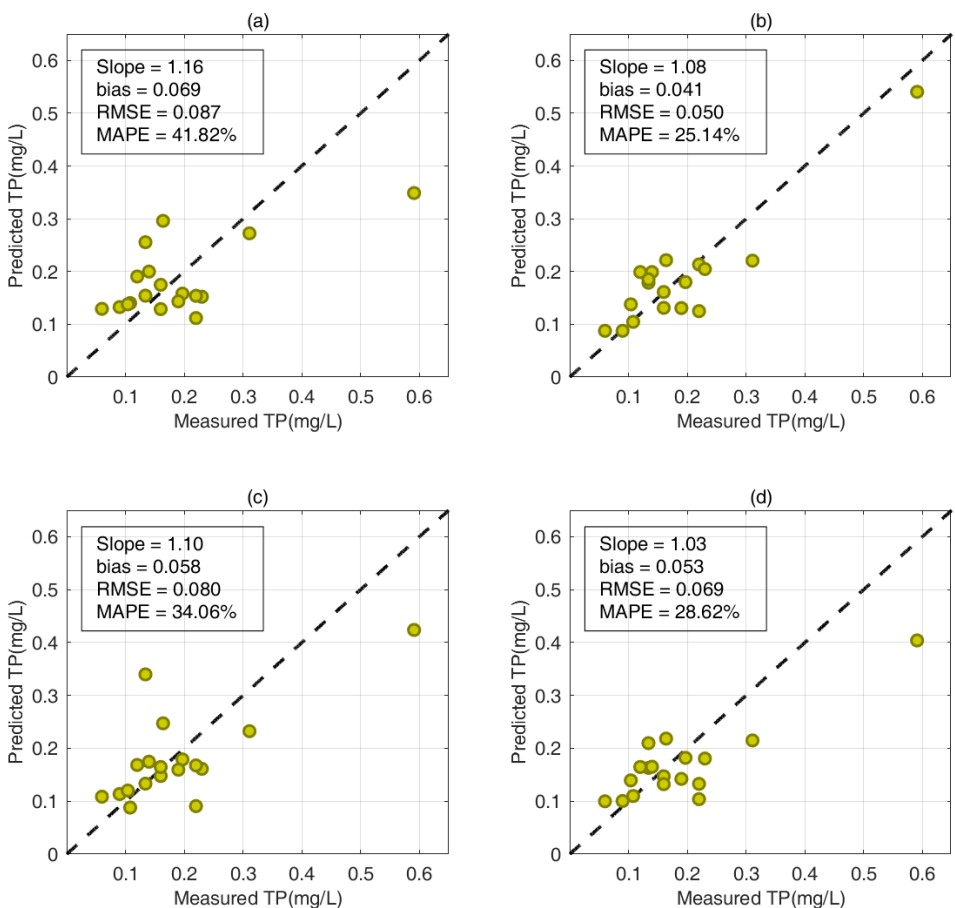

**Figure 5.** The scatter plot results of the predicted TP concentrations on the test set by four ensemble learning algorithms (AdaBoost (**a**), Catboost (**b**), RF (**c**), and XGBoost (**d**)), where the black dashed line represents the 1:1 line.

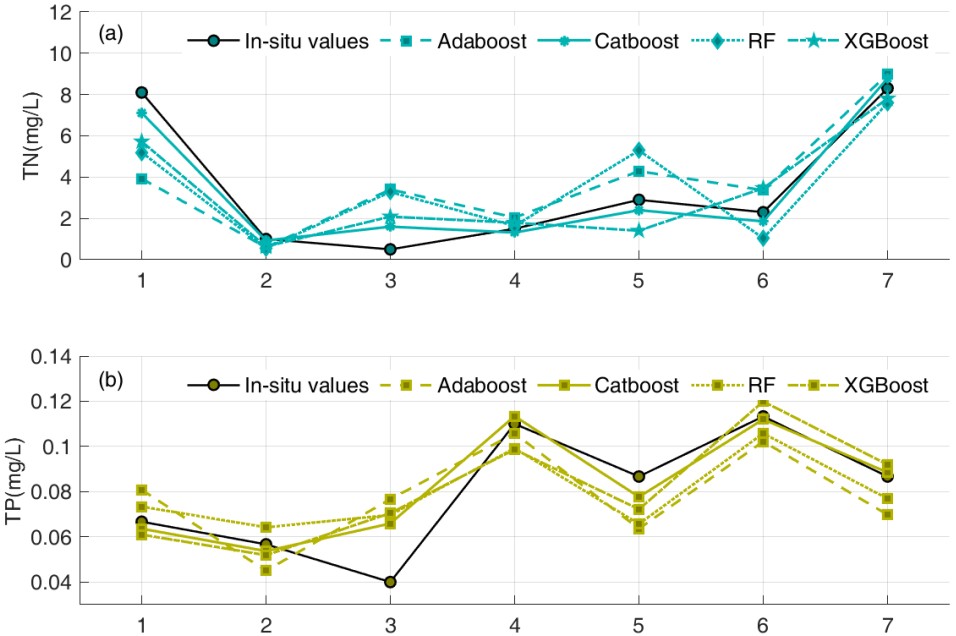

**Figure 6.** Comparison between predicted values obtained from reflectance at the image end and the measured (**a**) TN and (**b**) TP concentrations.

### 3.3. Results of UAV Inversion

The application of the optimal model (CatBoost) to the airborne hyperspectral images reveals the spatial distribution of TN and TP concentrations in Dianshan Lake, Yuandang Lake, and the inlet and outlet of Dianshan Lake. Figure 7 illustrates this distribution. In the northern and north–central regions of Dianshan Lake, the TN concentrations are relatively low, with a significant portion below 2 mg/L. In the southwest of Dianshan Lake and the eastern convergence area of Yuandang Lake, the TN concentrations fluctuate between 0 and 3 mg/L. In contrast, near the inlet and outlet of Dianshan Lake, particularly in the southern part of the central region, the TN concentrations are highest, exceeding 3 mg/L. In the western part of Yuandang Lake and the eastern part of Dianshan Lake, the TN concentrations are higher, ranging from approximately 3 to 4 mg/L. The overall TP concentration in Dianshan Lake remains between 0 and 0.25 mg/L, with lower concentrations in the open area of the lake centre, which were generally below 0.1 mg/L. The TP concentrations along the southeastern lake shoreline and near the river inlets and outlets are notably higher, especially in the river section between Jishui Port and Dianshan Lake, where the TP concentrations typically exceed 0.15 mg/L. At the confluence of Dianshan Lake and Yuandang Lake, the TP concentrations remain elevated, with overall higher TP concentrations observed in Yuandang Lake.

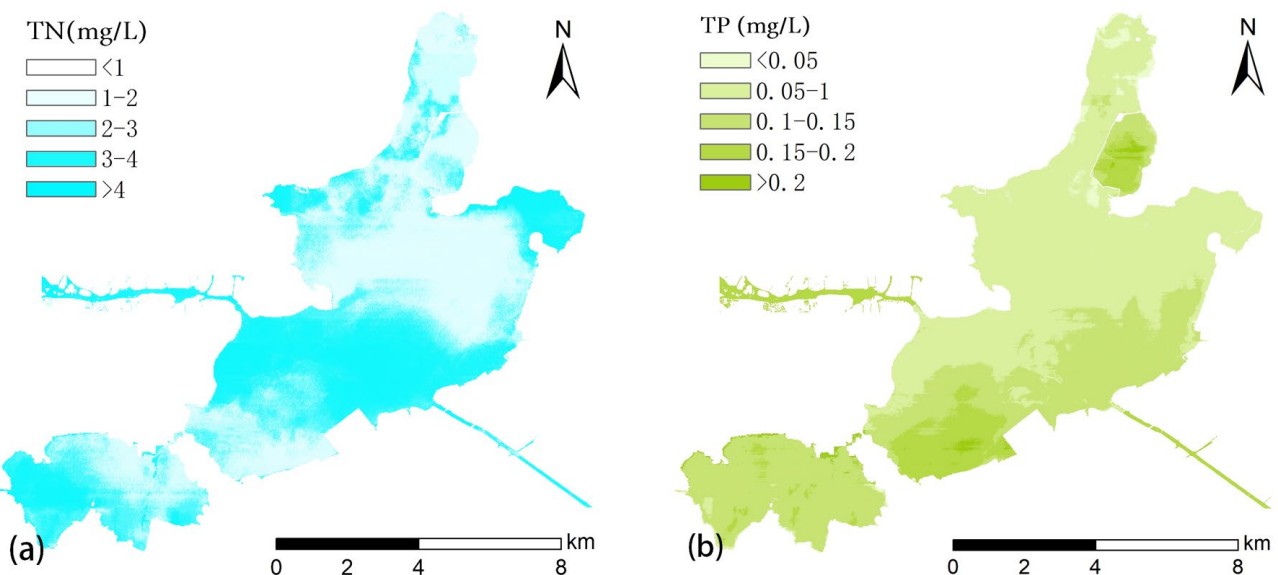

**Figure 7.** Thematic maps of (**a**) TN and (**b**) TP concentrations retrieved from airborne hyperspectral images.

### 4. Discussion

#### 4.1. Performance of Segmented Models

To investigate the superiority of seasonal models in the inversion of TN and TP, we utilised the complete dataset, Dataset 0, to establish an overall model. Subsequently, we built corresponding segmented models using Dataset 1 for the winter and spring seasons and Dataset 2 for the summer and autumn seasons to evaluate the performance of the overall model against segmented models. The RMSE and MAPE of the TN and TP models are shown in Figure 8 below. The results demonstrate that the segmented models outperform the overall model. As illustrated in Figure 8a, the RMSE of the TN and TP inversion results from the segmented models established by the four machine learning (ML) methods is consistently lower than that of the overall model. Figure 8b illustrates that, in terms of the MAPE metric, with the exception of the TN model constructed using AdaBoost and the TP model constructed using random forest, the segmented models established by the remaining three machine learning (ML) methods also exhibit lower errors. Therefore, it can be postulated that segmented models are more accurate in capturing the variations in TN and TP characteristics under different seasons, thereby improving the predictive perfor-

mance of the models. As previously stated in Section 1, TN and TP exhibit subtle variations in different seasons. Therefore, the adoption of segmented modelling approaches allows for a more comprehensive consideration of seasonal factors affecting nutrient concentrations in water bodies, thereby enhancing model adaptability and accuracy.

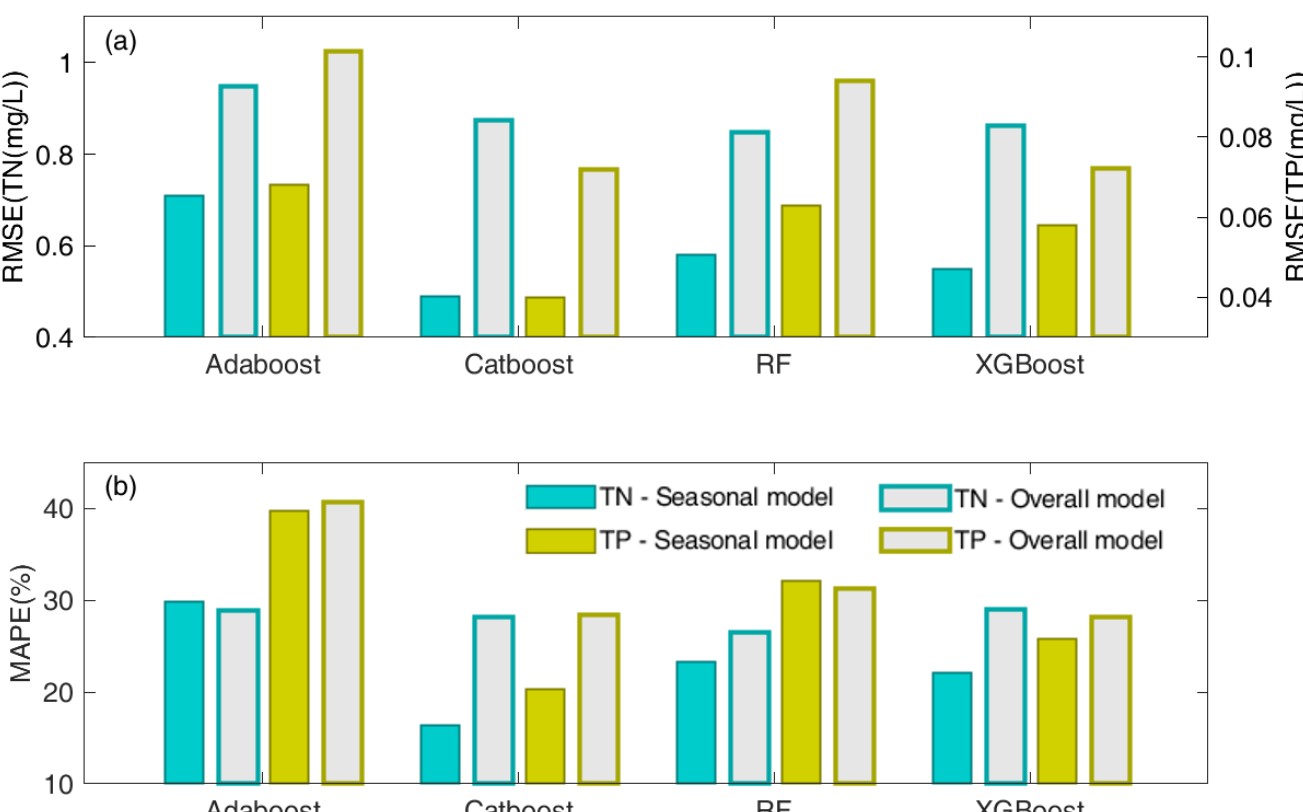

**Figure 8.** Bar charts of (**a**) RMSE and (**b**) MAPE for the overall model and seasonal models.

Moreover, it is important to highlight that the bar charts provide an intuitive demonstration of the performance of different machine learning algorithms. Figure 8 illustrates that, regardless of whether it is the overall model or the seasonal models, CatBoost demonstrates the smallest prediction errors for TN and TP. CatBoost is known for its ability to handle classification and regression tasks effectively. Given the constraints commonly encountered in water quality parameter data, such as the data acquisition costs and time limitations, the issue of small sample sizes is a common challenge [38]. In comparison to other traditional gradient boosting methods, CatBoost employs more sophisticated strategies to handle categorical features, rendering it more suitable for handling small samples and sparse data, thereby conferring it an advantage in predicting water quality parameters. In contrast to other ML algorithms, CatBoost exhibits superior training capabilities for small sample data, employing fully forgetting trees to better resist overfitting in limited data [33]. Furthermore, CatBoost incorporates adaptive learning rate adjustment and histogram-based optimisation, which facilitate more effective learning of data features and patterns during the training process. This, in turn, enhances model generalisation and predictive performance.

In conclusion, the application of seasonal modelling approaches and the CatBoost algorithm provides effective tools for gaining a deeper understanding of the spatiotemporal distribution of TN and TP in water bodies. These findings underscore the importance of utilising seasonal data and machine learning algorithms appropriately, thus offering robust support for better understanding and predicting changes in nutrient concentrations in water bodies.

### 4.2. Analysis of UAV Inversion Results

In this study, we observed that the distribution of algal blooms in the remote sensing maps of water quality parameters coincided with relatively high concentrations of TN and TP. We selected data from the boundary between Dianshan Lake and Yuandang Lake and plotted the distribution maps of algal blooms (Figure 1c shaded grey area), TN, and TP concentrations, as shown in Figure 9 below. It is evident that locations where algal blooms are detected typically exhibit higher concentrations of TN and TP. This observation suggests a significant correlation between the formation of algal blooms and the TN and TP contents in the water body, although further evidence is needed to establish the causality between the two. Firstly, high concentrations of TN and TP are commonly regarded as indicators of water eutrophication [22,46]. It is often observed that the growth of algae is significantly enhanced in eutrophic water bodies [47]. Consequently, the elevated concentrations of TN and TP observed in the areas affected by algal blooms may be indicative of an increase in the degree of water eutrophication, which in turn provides the optimal conditions for algal growth. Secondly, the presence of algal blooms may result in an increase in TN and TP concentrations in the water body. The growth of algae is dependent on the absorption of nitrogen and phosphorus nutrients from the surrounding water. Consequently, the proliferation of algae may result in an increase in the concentration of these nutrients within the water body. Therefore, the emergence of algal blooms may be attributed to the elevated levels of nitrogen and phosphorus within the water.

Previous studies have indicated that the primary sources of nutrients in lakes include external inputs, internal organic matter cycling, and sediment release [48]. External inputs may be a significant contributing factor to the elevated nitrogen and phosphorus concentrations observed at the inlets and outlets of lakes [49,50]. As illustrated in Figure 7 in Section 3.3, the concentration of TN at the inlet and outlet rivers of Dianshan Lake is notably high, while the concentration of TN in the lake body between these two rivers is also nearly at its highest levels. However, similar trends are not observed for the concentration of TP. To further investigate the influence of Jishui Port and Lanlu Port on the nitrogen and phosphorus concentrations in Dianshan Lake, spatial distribution maps of the TN and TP concentrations in the two rivers were plotted (Figure 10). The results indicate that the TN and TP concentrations are generally highest at Jishui Port, while the TN concentration is relatively high at Lanlu Port. However, the TP concentration shows a decreasing trend. This suggests that TN is influenced by river sources, possibly because the inlets and outlets of lakes serve as the primary channels for material exchange between lakes and their surrounding environments, thus they receive more significant nitrogen nutrient input from the surrounding areas. Furthermore, as illustrated in Figure 10a, the region surrounding Jishui Port is characterised by extensive areas of bare soil and agricultural land, where the application of agricultural fertilisers and pesticides, in conjunction with rainfall runoff, may facilitate the transport of nitrogen and phosphorus from the soil into the water body. Conversely, the area near Lanlu Port is predominantly forested, which may be associated with its lower TP concentration.

In future research, it is recommended that a more comprehensive approach be employed, including time series analysis, field sampling, and laboratory experiments, in order to further investigate the relationship between the TN and TP concentrations in lakes, algal bloom distribution, and external inputs from rivers. It is also worth noting that although the model was applied only to summer imagery, the seasonal model can be applied to all periods. In the future, it would be beneficial to consider incorporating data from the winter season in order to achieve a more comprehensive spatial distribution analysis. This would involve comparing the characteristics and trends of TN and TP across different seasons. This would contribute to a more comprehensive understanding of the dynamic processes of lake ecosystems, providing deeper and more comprehensive scientific foundations for the management of lake eutrophication and environmental protection.

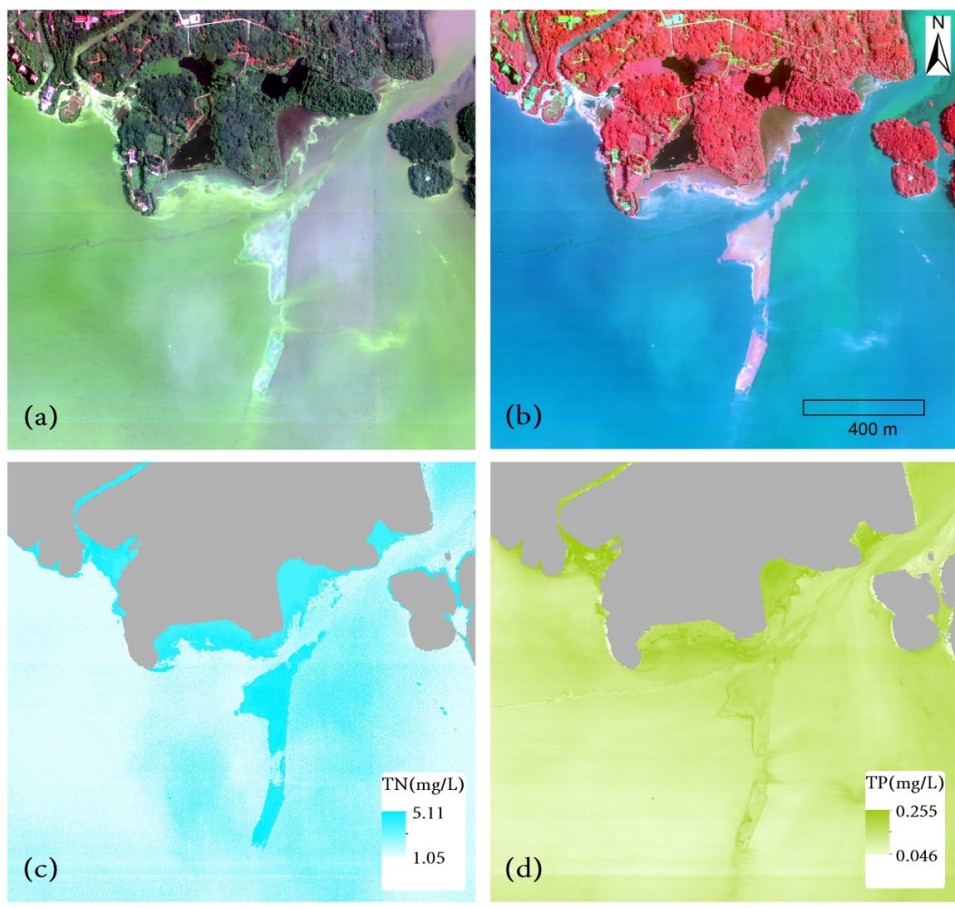

**Figure 9.** (**a**) RGB true-colour image, (**b**) false-colour image, (**c**) spatial distribution map of TN, and (**d**) TP concentrations at locations of algal bloom distribution.

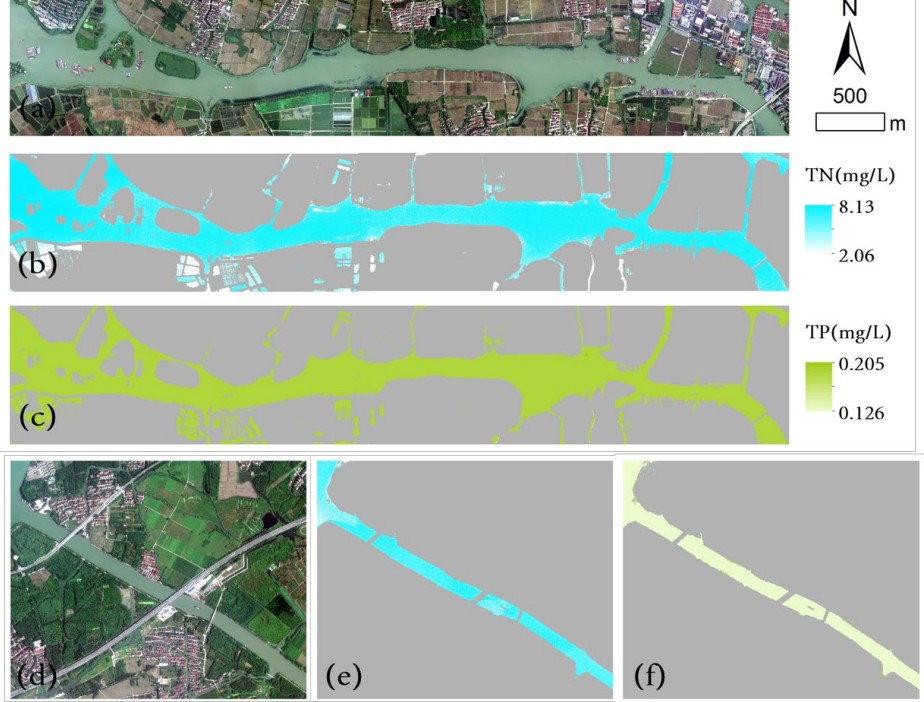

**Figure 10.** (**a**,**d**) RGB true-colour images of the inlet river (Jishui Port) and outlet river (Lanlu Port) of Dianshan Lake and (**b**,**e**) spatial distribution maps of TN and (**c**,**f**) TP concentrations.

*4.3. Strengths and Limitations*

Currently, numerous water colour satellite sensors have been developed for open water bodies such as oceans. However, they typically possess lower resolutions, rendering them unsuitable for inland small water bodies. Furthermore, there are currently no specialised satellite sensors tailored for the optical properties of more complex Type II water bodies [6]. Although the spatial resolution of land observation satellites may be sufficient, they lack the required spectral bands sensitive to water quality. With the rapid advancement of airborne remote sensing technology, airborne hyperspectral imaging technology is expected to become increasingly mature in the future, providing a powerful approach for monitoring the water quality of small-to-medium-sized lakes and urban rivers. The continuous progress of this technology will make water quality monitoring more precise and efficient, offering reliable data support for water resource management and environmental protection. The near-surface remote sensing imagery data employed in this study feature high spatial and spectral resolutions, offering flexible image acquisition times and proximity to the water surface, thereby simplifying atmospheric correction processes. Consequently, the study demonstrates the applicability of near-surface high spectral remote sensing data in inland small lakes and rivers. Furthermore, with regard to the non-optically active water quality parameters TN and TP, the dataset was divided by season in order to identify differences in reflectance between the summer and autumn seasons. This led to the development of relative overall models for TN and TP. These models exhibit higher precision, employing CatBoost seasonal models, and were successfully applied to Dianshan Lake, Yuandang Lake, Jishui Port, and Lanlu Port. The study provides a crucial reference value for small sample modelling. Furthermore, in comparison to the overall model, it can more accurately track the seasonal distribution of TN and TP. By accurately monitoring the seasonal changes of these water quality parameters, we can gain a deeper understanding of the ecological and health status of water bodies, which in turn can aid in predicting the risk of eutrophication and taking measures to mitigate or prevent water pollution. Furthermore, these data can be employed to assess the efficacy of management strategies and to monitor the implementation of environmental policies. This provides a foundation for the continuous improvement of water quality.

While the study has made commendable progress, it is essential to address several noteworthy issues. Firstly, the study only utilised four ensemble ML methods. Currently, deep neural network algorithms such as CNN and DNN are increasingly prominent in water quality detection [51–53]. Due to the limitations of the data sample size, deep learning algorithms were not employed in this study. Future considerations could involve increasing the data volume to incorporate more methods for comparison and attain optimal results. Secondly, the models were only applied to imagery from a single date, and the results of the inverse performance may not be representative for the study of the water quality status of the whole lake. Furthermore, the airborne hyperspectral data employed in our study, due to its high spectral complexity and large volume, has not been publicly shared. Consequently, the availability of data and the possibility of external validation for this research are limited. Given the flexibility of near-surface image acquisition, future endeavours could involve obtaining imagery data from additional seasons in order to achieve a more comprehensive temporal and spatial distribution analysis of TN and TP.

## 5. Conclusions

This study employed a remote sensing estimation method to determine the concentrations of TN and TP in Lake Dianshan and its main inflowing and outflowing rivers. The study used airborne hyperspectral remote sensing image data and a seasonal modelling approach. The reflectance and measured TN and TP data from different seasons were analysed and differences in reflectance and the TN and TP concentrations were identified between the summer–autumn and spring–winter seasons. The CatBoost-method-based seasonal models demonstrated superior predictive capabilities for the TN and TP concentrations, exhibiting significantly higher accuracy compared to other ensemble learning models.

Application of the CatBoost model to images captured on 15 June 2022 yielded TN and TP concentration maps, which revealed spatial distributions influenced by external inputs and closely associated with algal blooms. Further analysis indicated that external inputs from rivers may be a significant contributor to elevated nitrogen and phosphorus concentrations at lake inlets. In conclusion, this study demonstrated the feasibility of airborne remote sensing images and CatBoost seasonal retrieval methods for monitoring the water quality in small inland water bodies. It provided an important reference value for the management of inland water bodies. Future research could apply the model to more dates of airborne remote sensing image data to further explore the inherent relationships between lake TN and TP concentrations, algal bloom distributions, and river external inputs. This would provide a more in-depth and comprehensive scientific basis for lake management and environmental protection.

**Author Contributions:** Conceptualisation, L.D.; methodology, Y.H. and L.D.; validation, L.L., Z.Y. and C.G.; resources, D.H., X.W. and Y.W.; data curation, C.G. and L.L.; writing—original draft preparation, L.D.; writing—review and editing, L.D., Z.Y. and C.G.; supervision, Z.Y.; project administration, C.G., L.L. and Z.Y.; funding acquisition, Y.W. and D.H. All authors have read and agreed to the published version of the manuscript.

**Funding:** This research was funded by the National Key R&D Program of China (Grant No. 2022YFB3902000): Very long wavelength infrared (VLWIR) hyperspectral imaging technology, the National Civil Aerospace Project of China (No. D040102), the Shanghai 2021 "Science and Technology Innovation Action Plan" Social Development Science and Technology Research Project (21DZ1202500), the Jiangsu Provincial Water Conservancy Science and Technology Research Project (No. 2020068), and the Science and Technology Project of the Shanghai Municipal Water Bureau (Shanghai Branch 2021-10, Shanghai Branch 2018-07).

**Data Availability Statement:** The original contributions presented in the study are included in the article, further inquiries can be directed to the corresponding author/s.

**Acknowledgments:** We would like to thank the Shanghai Natural Resource Satellite Application Technology Centre for providing validation data. We also extend our gratitude to all contributors to this paper, as well as to all anonymous reviewers for their constructive comments on the study.

**Conflicts of Interest:** The authors declare no conflicts of interest.

## Appendix A

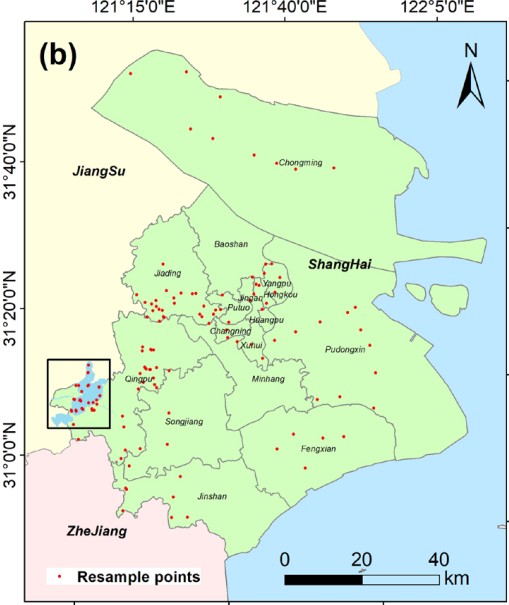

**Figure A1.** A larger view of Figure 1b.

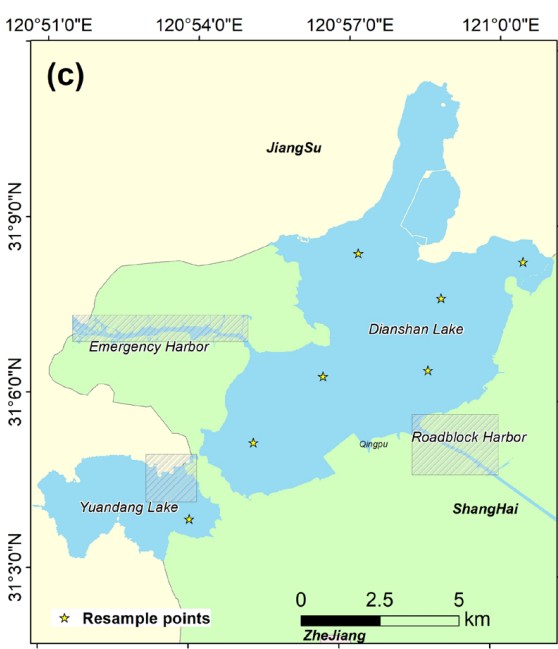

**Figure A2.** A larger view of Figure 1c.

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
