# Peer review of "Seasonal Monitoring Method for TN and TP Based on Airborne Hyperspectral Remote Sensing Images"

_remotesensing, doi:10.3390/rs16091614_

Round 1

Reviewer 1 Report

Comments and Suggestions for Authors

This paper proposes a seasonal monitoring method for TN and TP based on proximal hyperspectral remote sensing images.

The detailed comments are summarized as follows:

1. In the introduction section, the authors only incorporate the main objective of this study. The motivations and main contributions of this paper should be incorporated.

2. The proposed method can conduct feature selection from remote sensing data. To my knowledge, remote sensing images can be used for providing rich information for analysis. However, the captured sensing images may be affected the inclement weather, which will be degraded. Therefore, several image enhancement, restoration and denoising methods [1-5] can be considered as pre-processing techniques to improve the quality of these captured images.

1) A Novel Truncated Norm Regularization Method for Multi-channel Color Image Denoising

2) Multi-purpose Oriented Single Nighttime Image Haze Removal Based on Unified Variational Retinex Model

3) Nighthazeformer: Single nighttime haze removal using prior query transformer

4) Degradation-Adaptive Neural Network for Jointly Single Image Dehazing and Desnowing

5) Visibility restoration for real-world hazy images via improved physical model and Gaussian total variation

3. Beside RMSE and MAPE, could you provide more accuracy assessment metrics for evaluations.

4. The authors employ ensemble learning strategies, such as Adaboost, CatBoost, Random Forest and XGBoost. Which is the best performance for these selected learning strategies? Why not use some recent CNN-based or Transformer-based learning techniques?

5. In the discussion section, the authors should discuss the limitations of the proposed method. Meanwhile, the potential solutions should be given for these limitations.

Based on the detailed comments, I recommend major revisions for this manuscript. I believe that the revised manuscript can be accepted and I hope that the authors should carefully revise this manuscript according to my comments.

Comments on the Quality of English Language

Moderate editing of English language required

Author Response

Dear Reviewer,

Hello! Thank you for carefully reviewing our manuscript and providing valuable feedback. We have revised the original manuscript based on your comments. Attached is our response to your feedback. Thank you for your consideration!

Best regards

Reviewer 2 Report

Comments and Suggestions for Authors

General comments

Dong et al. aim to examine remote sensing estimating techniques for determining the concentrations of total nitrogen (TN) and total phosphorus (TP) in Lake Dianshan and the rivers that surround it. The study also reveals the seasonal fluctuations in these concentrations and highlights the superior performance of Random Forest (RF) models in comparison to other methods. The debate focuses on the regional distribution of TN and TP concentrations in response to external inputs and algal blooms, providing vital insights into the dynamics of water quality. It highlights the potential of the study as a reference for future research on employing proximal hyperspectral remote sensing pictures to analyze small water bodies. However, feature selection-based classification among TN, TP, and normal water samples can enhance the proper visibility of the data and make it easier for readers. Furthermore, considering the results of this study, some future implications will be added.

Specific comments

Revise the title

L17 & 18: Give space between abbreviations, also for the whole manuscript

L40: Give space between text and references, also for the whole manuscript

L116: Give reference

L125: Remove “we”, also for the whole manuscript

L126: Add reflectance range

Figure 2: Dataset2 have higher reflectance, while in Datset1 the average reflectance is higher, please recheck these reflectance patterns

L154: Can we name the UAV Airborne imaging spectrometer as proximal hyperspectral remote sensing data?

Figure 3: Increase the image quality

Comments on the Quality of English Language

The English language is fine

Author Response

(The authors gave the same response as above.)

Reviewer 3 Report

Comments and Suggestions for Authors

First, I would like to tell that the paper shows several strong points that stand out. First, it introduces an innovative methodology for seasonal tracking of Total Nitrogen (TN) and Total Phosphorus (TP) levels using proximal hyperspectral remote sensing images. This approach addresses a significant challenge in environmental tracking and offers potential benefits for resource management and ecosystem preservation. I would like to remark that the detailed description of the method enhances reproducibility and helps with adoption by other researchers and practitioners. Congratulations!

The paper includes a comprehensive analysis of seasonal variations in TN and TP levels, providing valuable insights into temporal trends and environmental dynamics. This seasonal tracking capability is essential for understanding ecosystem dynamics, identifying long-term trends, and informing management strategies to mitigate nutrient pollution and preserve water quality.

Still, I find parts that should be improved:

·         Clarity and Organization: The paper would benefit from improved clarity and organization, particularly in the presentation of the methodology and results sections. Please include clear and concise descriptions of data processing steps, feature extraction methods, and model implementation details; this would enhance the readability and understanding of the paper.

·         Discussion of Practical Implications: In addition to describing the technical parts of the proposed method, the paper should discuss the practical implications and potential applications of seasonal tracking for TN and TP levels. Exploring how the proposed approach could inform decision-making, guide management practices, and support environmental policy initiatives, from my viewpoint, it would increase the relevance of the research.

·         Future Research Directions: Including a section on future research directions would enrich the paper's contribution by identifying potential avenues for further investigation and improvement. I.E. you could discuss unresolved challenges, emerging technologies, and opportunities for innovation in environmental tracking.

·         Data Availability and Accessibility: The paper should provide more information about the availability and accessibility of the hyperspectral remote sensing data used in the study. Transparency regarding data sources, acquisition protocols, and potential restrictions or limitations on data access would enhance the reproducibility of the research and help with future studies building on this work.

·         Model Interpretability: While the paper discusses the methodology for TN and TP estimation using hyperspectral data, there is limited discussion on the interpretability of the models or the spectral features driving the predictions. Providing insights into the relationship between spectral signatures and environmental parameters would improve the understanding of the tracking system.

Finally, I would like to say that the paper presents a very promising method for seasonal tracking of TN and TP levels using proximal hyperspectral remote sensing images. Addressing the identified weaknesses and incorporating the suggested improvements would strengthen the paper's contribution and help with its adoption by the scientific community for environmental tracking and management applications.

Author Response

Thank you very much for taking the time to review our manuscript. Attached is our detailed response to your comments. We appreciate your attention!

Round 2

Reviewer 1 Report

Comments and Suggestions for Authors

After carefully reviewing the revised manuscript, there still exist several concern to be addressed.

1. In Table 4, the authors only compare some machine learning based models. Could you provide more comparisons with deep learning based models?

2. In the introduction section, the cited references may be out of date. The authors should cite more latest references.

3. Some figures are not clear. The authors should improve the quality of these figures.

4. What are the differences between the data through aerial photography and images? I think the captured remote sensing images may include more information. Could you in detail explain the differences and conduct the experiments to demonstrate the advantages of your data? I hope that the authors can provide experiments to demonstrate the superiority of the proposed method. To my knowledge, remote sensing data incorporate more rich information.

Comments on the Quality of English Language

Extensive editing of English language required

Author Response

Dear reviewer 1

Greetings! Thank you for your careful review of our manuscript and your valuable comments. Based on your comments, we have revised the original manuscript. Our response to your feedback is attached. Thank you for your viewing!

With best regards

Reviewer 2 Report

Comments and Suggestions for Authors

The authors have well revised the manuscript. 

Author Response

Dear reviewer 2

Greetings! Thank you for carefully reviewing our manuscript and providing your valuable responses. 

With best regards

Reviewer 3 Report

Comments and Suggestions for Authors

Thank you for your response to my review and for the revisions made to the manuscript. I appreciate the effort you've put into addressing the comments and suggestions provided. The revisions have significantly enhanced the clarity, relevance, and rigor of the research. Below, I provide further comments on the new manuscript:

The revisions to sections 2.3.2 and 3.1 have improved the clarity of the method and results sections. The descriptions of data processing steps, feature extraction methods, and model implementation details are now more concise and understandable.

Adding the new subsection "4.3. Strengths and Limitations" enhances the relevance of the research and underscores its potential impact on decision-making and environmental policy initiatives.

While it's understandable that the hyperspectral remote sensing data used in the study may not be publicly available due to resource and technical constraints, the transparency provided regarding data acquisition equipment and processing methods in Section 2.2.3 is commendable.

The additional insights into the relationship between spectral features and environmental parameters, particularly TN and TP concentrations, provide a deeper understanding of the tracking system.

I have no further major concerns regarding the manuscript and recommend its acceptance for publication pending minor revisions.

Below, I outline two minor revisions that would further enhance the clarity and completeness of the manuscript:

·         I suggest providing a brief summary or conclusion at the end of the "Model Interpretability" section to tie together the discussion points and reinforce the importance of understanding these relationships for the tracking system.

·         Considering the constraints on sharing the hyperspectral remote sensing data used in the study, I recommend briefly acknowledging this limitation in the manuscript's limitations section to provide transparency

Congratulations for the work!

Kind regards,

Author Response

Dear reviewer 3

Greetings! Thank you for your careful review of our manuscript and your valuable comments. Based on your comments, we have revised the original manuscript. Our response to your feedback is attached. In addition, we have touched up the full text in the manuscript in English language this time, thank you for viewing!

With best regards
